

# Transcranial magnetic stimulation enhances the specificity of multiple sclerosis diagnostic criteria: a critical narrative review

Nicholas J. Snow[1], Hannah M. Murphy[1], Arthur R. Chaves[2,3,4], Craig S. Moore[1] and Michelle Ploughman[1]

[1] Faculty of Medicine, Memorial University of Newfoundland, St. John's, NL, Canada
[2] Faculty of Health Sciences, Interdisciplinary School of Health Sciences, University of Ottawa, Ottawa, ON, Canada
[3] Neuromodulation Research Clinic, The Royal's Institute of Mental Health Research, Ottawa, ON, Canada
[4] Département de Psychoéducation et de Psychologie, Université du Québec en Outaouais, Gatineau, QC, Canada

Corresponding author
Nicholas J. Snow, njsnow@mun.ca

## ABSTRACT

**Background:** Multiple sclerosis (MS) is an immune-mediated neurodegenerative disease that involves attacks of inflammatory demyelination and axonal damage, with variable but continuous disability accumulation. Transcranial magnetic stimulation (TMS) is a noninvasive method to characterize conduction loss and axonal damage in the corticospinal tract. TMS as a technique provides indices of corticospinal tract function that may serve as putative MS biomarkers. To date, no reviews have directly addressed the diagnostic performance of TMS in MS. The authors aimed to conduct a critical narrative review on the diagnostic performance of TMS in MS.

**Methods:** The authors searched the Embase, PubMed, Scopus, and Web of Science databases for studies that reported the sensitivity and/or specificity of any reported TMS technique compared to established clinical MS diagnostic criteria. Studies were summarized and critically appraised for their quality and validity.

**Results:** Seventeen of 1,073 records were included for data extraction and critical appraisal. Markers of demyelination and axonal damage—most notably, central motor conduction time (CMCT)—were specific, but not sensitive, for MS. Thirteen (76%), two (12%), and two (12%) studies exhibited high, unclear, and low risk of bias, respectively. No study demonstrated validity for TMS techniques as diagnostic biomarkers in MS.

**Conclusions:** CMCT has the potential to: (1) enhance the specificity of clinical MS diagnostic criteria by "ruling in" true-positives, or (2) revise a diagnosis from relapsing to progressive forms of MS. However, there is presently insufficient high-quality evidence to recommend any TMS technique in the diagnostic algorithm for MS.

## INTRODUCTION

Multiple sclerosis (MS) is an immune-mediated neurodegenerative and neuroinflammatory disease characterized by chronic central nervous system (CNS) degeneration with intermittent attacks of inflammatory demyelination and axonal damage (*Reich, Lucchinetti & Calabresi, 2018*). Mitigation of disease activity, disease progression, and disability accumulation requires early and correct diagnosis (*McNicholas et al., 2018*). To diagnose MS in a patient with a history suggestive of a demyelinating episode, clinicians must find evidence of lesion dissemination in space and time (*Poser et al., 1983*). In the 2017 McDonald criteria (*Thompson et al., 2018*), magnetic resonance imaging (MRI) and cerebrospinal fluid (CSF) oligoclonal bands aid the clinical history and exam in finding these features. The 2017 McDonald criteria are highly sensitive; however, their low specificity can lead to misdiagnosis, thus resulting in delayed diagnosis and unnecessary treatment in some individuals (*Filippi et al., 2022*; *Gobbin et al., 2019*; *McNicholas et al., 2018*). As such, it is desirable to discover biological markers (biomarkers) of disease activity in MS that have the sensitivity to identify subclinical lesions early in the disease course, while possessing high specificity for MS-related disease processes (*Bielekova & Martin, 2004*).

A biomarker is "an objectively measured indicator of normal biological processes, pathogenic processes, or … responses to a therapeutic intervention" (*Bielekova & Martin, 2004*). Biomarkers can aid diagnosis, classify the extent of disease, observe natural history, or monitor responses to treatments (*Atkinson et al., 2001*). The diagnostic utility of a biomarker is based on its performance against a reference standard (*e.g.*, clinical-radiologic diagnostic criteria, histopathological diagnosis) (*Adeniyi et al., 2016*). A sensitive biomarker is one that yields a positive or abnormal result in a high proportion of individuals who have the disease (*Adeniyi et al., 2016*). A specific biomarker has a normal or negative result in a high proportion of individuals without the disease (*Adeniyi et al., 2016*). A diagnostic biomarker should ideally have both high sensitivity and high specificity (*Adeniyi et al., 2016*). Biomarkers are of interest in the broader biomedical literature because they can offer objective, biologically plausible information about a disease process that may go undetected by a patient (*Strimbu & Tavel, 2010*). In some cases, biomarker-based findings can precede clinical endpoints throughout the disease's natural history, leading to earlier diagnosis or signifying a change in the disease course (*Andersen et al., 2021*). In other cases, biomarker results can help distinguish a disease from other entities, leading to the correct diagnosis and targeted management (*Hayes, 2015*). In MS, a hypothetical diagnostic biomarker could be valuable to narrow the differential diagnosis in a patient with undifferentiated lesions on neuroimaging, or arrive at an earlier diagnosis in a patient with signs and symptoms suggestive of a demyelinating event (*Bielekova & Martin, 2004*; *Paul, Comabella & Gandhi, 2019*).

Transcranial magnetic stimulation (TMS) measurements are putative biomarkers for MS diagnosis and monitoring (*Alsharidah et al., 2022*; *Simpson & Macdonell, 2015*). Briefly, TMS uses a time-varying magnetic field to induce an electric field that is parallel to the surface of the brain (*Siebner et al., 2022*). Depending on TMS coil architecture and

orientation, pulse waveform, and stimulation intensity, the TMS-induced electric field can produce an electric current that preferentially depolarizes myelinated axons of superficial presynaptic layer II/III/V cortical pyramidal neurons in the precentral gyrus (*Rossini et al., 2015*; *Siebner et al., 2022*; *Spampinato et al., 2023*). The cortical pyramidal neuron action potential activates spinal corticospinal tract axons either directly or *via* mono- or polysynaptic inputs involving both inter- and intracortical connections (*Kesselheim et al., 2023*; *Siebner et al., 2022*; *Spampinato et al., 2023*). The corticospinal tract volley leads to activation of the spinal nerves, peripheral nerve(s), and motor units corresponding to the target muscle, eliciting a characteristic deflection in the surface electromyography (EMG) trace—the motor evoked potential (MEP) (*Groppa et al., 2012*; *Rossini et al., 2015*). The amplitude, latency, morphology, and conditioned responses of MEPs (Fig. 1) reflect the activity and function of corticospinal pyramidal neurons and CNS interneurons in relation to motor output (Table 1) (*Groppa et al., 2012*; *Rossini et al., 2015*; *Ziemann et al., 2015*). Various TMS measures can characterize CNS demyelination, axonal damage, and/or excitotoxicity in MS (*Chen et al., 2008*; *Snow et al., 2019*; *Stampanoni Bassi et al., 2020*; *Vucic et al., 2023*). For example, CNS demyelination, axonal damage, and excitotoxicity can be detected in the MEP waveform as prolonged MEP onset latency or MEP dispersion, reduced MEP amplitude, and shortened post-MEP corticospinal silent period (CSP), respectively (*Fernández et al., 2013*; *Snow et al., 2019*; *Stampanoni Bassi et al., 2020*) (Fig. 1).

To aid diagnosis, a biomarker should balance the probability that a patient has MS and does not have an alternative diagnosis (*Adeniyi et al., 2016*; *Atkinson et al., 2001*; *Richardson & Wilson, 2015*). This problem is relevant in MS because the differential diagnosis is broad (*Solomon, 2019*; *Solomon et al., 2023*; *Wildner, Stasiolek & Matysiak, 2020*) and current diagnostic criteria—the 2017 McDonald criteria (*Thompson et al., 2018*)—are sensitive but not specific (*Filippi et al., 2018*, *2022*; *Gobbin et al., 2019*; *van der Vuurst de Vries et al., 2018*), resulting in a high rate of misdiagnosis (*Dixon & Robertson, 2018*; *Solomon, 2019*; *Solomon, Naismith & Cross, 2019*). A biomarker should likewise be reliable and valid; biologically plausible and clinically relevant; and practical and cost-effective (*Adeniyi et al., 2016*; *Atkinson et al., 2001*; *Bielekova & Martin, 2004*). A previous systematic review of TMS biomarker studies in MS, by this research group (*Snow et al., 2019*), highlighted cross-sectional relationships between various TMS techniques and MS clinical outcomes. However, the previous review did not directly address the role of TMS in MS diagnosis (*Snow et al., 2019*). Thus, the current critical narrative review aimed to explore the diagnostic accuracy of TMS techniques in MS.

## SURVEY METHODOLOGY
This review followed the SANRA checklist (*Baethge, Goldbeck-Wood & Mertens, 2019*). The review is intended for clinicians and researchers with an interest in MS neurophysiology.

### Search strategy
The search was planned by the entire study team and performed by a single author (NJS).

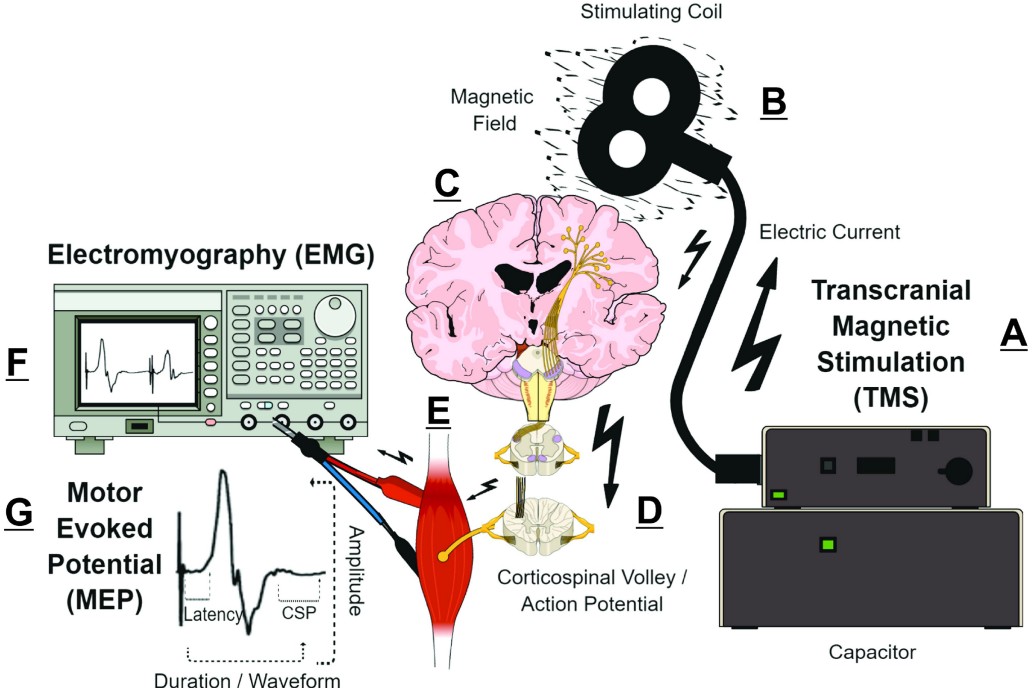

**Figure 1** **Simplified schematic of transcranial magnetic stimulation (TMS)-induced generation of motor evoked potential (MEP).** (A) Pulse generator produces an electric current that is stored in, and rapidly discharged from, a large capacitor into the stimulating coil. (B) The insulated coil contains windings of copper, which generate a focal magnetic field from the electric current. (C) The magnetic field undergoes little attenuation from extracerebral tissues and painlessly induces an electric current in underlying layer II/III/V pyramidal neuron axons at the gyral crown of the primary motor cortex. (D) Activation of corticospinal pyramidal neurons elicits descending corticospinal volleys from the brain to the spinal cord by directly activating pyramidal tract neurons, or indirectly *via* interneurons that synapse onto the pyramidal tract/lateral corticospinal tract. (E) The descending corticospinal volley activates the target muscle, *via* stimulation of anterior horn cells, peripheral nerve, and motor unit. (F) The TMS-induces motor evoked potential (MEP) can be recorded *via* electromyography (EMG), with recording electrodes placed over the belly of the target muscle. (G) Analyzing the amplitude, latency, duration, and waveform characteristics provides information on the excitability and conduction characteristics of corticospinal pyramidal cells; the post-MEP corticospinal silent period (CSP) characterizes the excitability of corticospinal inhibitory interneurons. See refs: (*Chaves et al., 2021*; *Rossini et al., 2015*; *Siebner et al., 2022*; *Snow et al., 2019*; *Spampinato et al., 2023*) .

A single author (NJS) searched the PubMED, Embase, Web of Science, and Scopus electronic databases for studies published between January 1, 1985 (the first year of TMS publication) (*Barker, Jalinous & Freeston, 1985*) and February 28, 2022. The following search terms were adapted for each database:

("multiple sclerosis" (all fields) OR "clinically isolated syndrome" (all fields)) AND ("transcranial magnetic stimulation" (all fields)) AND (sensitiv* (all fields) OR specific* (all fields) OR "predictive value" (all fields) OR "likelihood ratio" (all fields) OR "odds ratio" (all fields) OR "risk ratio" (all fields) OR "hazard ratio" (all fields)).

A single author (NJS) also scanned reference lists of relevant review articles and full-text articles.

**Table 1 Description of transcranial magnetic stimulation (TMS) outcomes.**

| TMS outcome | Stimulation characteristics | Mechanism of action | Studies |
|---|---|---|---|
| *Motor thresholds* | | | |
| Resting motor threshold (RMT) | Lowest TMS stimulus intensity to elicit MEP with peak-to-peak amplitude of 50 μV in at least five of 10 consecutive trials, in resting target muscle (*Rossini et al., 2015*). Reported as % MSO. | Reflects the strength and size of the most excitable elements of the target muscle cortical representation, activity of glutamate its receptors (*e.g.*, AMPA), and function of ion channels (*e.g.*, VGSC) in cortical and spinal neuron populations (*Rossini et al., 2015*; *Ziemann et al., 2015*). Indicates the bias level of the cortical representation (*Groppa et al., 2012*; *Rossini et al., 2015*). May index demyelination and axonal damage (*Snow et al., 2019*). | *Cruz-Martínez et al. (2000)*, *Schmierer et al. (2002)* |
| *Corticospinal excitability* | | | |
| Motor evoked potential (MEP) | Deflection in EMG trace of target muscle following delivery of threshold or suprathreshold TMS pulse to target muscle cortical representation (*Rossini et al., 2015*; *Ziemann et al., 2015*). Measured in active or resting muscle. MEP amplitude increases in sigmoidal relationship with TMS stimulus intensity. This stimulus-response curve requires incrementally increasing TMS stimulus intensity to examine corresponding increases in MEP amplitudes due to faster temporo-spatial summation at cortico-motoneuronal synapses (*Rossini et al., 2015*). Higher stimulus intensities improve synchronization of neuronal firing (*Magistris et al., 1998*). The stimulus-response curve indexes the excitability of the least to most excitable neuronal populations in the motor representation (*Groppa et al., 2012*; *Ridding & Rothwell, 1997*). Corticospinal conduction properties can be examined by observing MEP latency or waveform characteristics (*Groppa et al., 2012*; *Rossini et al., 2015*; *Snow et al., 2019*). | Reflects summation of action potentials in corticospinal axons which synapse on spinal motor neurons. MEP amplitudes and stimulus-response curves characterize the recruitment gain, variability, and excitability of corticospinal neuron populations (*Capaday, 1997*; *Carson et al., 2013*; *Devanne, Lavoie & Capaday, 1997*; *Ridding & Rothwell, 1997*; *Talelli et al., 2008*). Reflects activity of glutamatergic, GABAergic, and putatively serotonergic and noradrenergic neurons (*Rossini et al., 2015*; *Ziemann et al., 2015*). May index demyelination-induced conduction deficits or axonal damage (*Snow et al., 2019*). | *Cruz-Martínez et al. (2000)*, *Hess et al. (1987)*, *Kale et al. (2009)*, *Kale, Agaoglu & Tanik (2010)*, *Kandler et al. (1991)*, *Mayr et al. (1991)*, *Pisa et al. (2020)*, *Ravnborg et al. (1992)*, *Schmierer et al. (2000)*, *Tataroglu et al. (2003)* |
| *Corticospinal conduction* | | | |
| Central motor conduction time (CMCT) | Difference between motor cortex-to-muscle latency (onset latency of MEP) and spinal cord-/brainstem-to-muscle latency (*Rossini et al., 2015*). Spinal cord-/brainstem-to-muscle latency is estimated by stimulating spinal nerve roots (nerve root latency) or the peripheral nerve (F-wave latency) innervating the target muscle (*Rossini et al., 2015*). Measured in active or resting muscle. Reported as difference between motor cortex-to-muscle and spinal cord-/brainstem-to-muscle latencies. | Reflects cortical output latency, the conduction time of the corticospinal tract between the motor cortex and brainstem or spinal motor neurons (*Rossini et al., 2015*). Posited as one of the more clinically useful TMS methods in examinations of MS because of its ability to detect demyelination and conduction loss (*Chen et al., 2008*; *Vucic et al., 2023*) | *Beer, Rösler & Hess (1995)*, *Caramia et al. (2004)*, *Cruz-Martínez et al. (2000)*, *Facchetti et al. (1997)*, *Hess et al. (1987)*, *Jung et al. (2006)*, *Kale et al. (2009)*, *Kale, Agaoglu & Tanik (2010)*, *Kandler et al. (1991)*, *Leocani et al. (2006)*, *Magistris et al. (1999)*, *Mayr et al. (1991)*, *Ravnborg et al. (1992)*, *Schmierer et al. (2000, 2002)*, *Tataroglu et al. (2003)* |

(Continued)

| TMS outcome | Stimulation characteristics | Mechanism of action | Studies |
|---|---|---|---|
| Triple stimulation technique (TST) | Delivery of suprathreshold TMS over the target muscle cortical representation, supramaximal electrical stimulation over the distal part of the peripheral nerve supplying the target muscle, and a second supramaximal electrical stimulation over the proximal part of the same nerve (Erb's point) (*Rossini et al., 2015*). Timing of stimuli is individualized to ensure action potentials induced by TMS collide with the corticospinal volleys from peripheral nerve stimulations (*Rossini et al., 2015*). A TST test curve is compared to a control curve derived from triple stimulation of the peripheral neve (*Rossini et al., 2015*). Reported as amplitude/area ratio of test curve relative to control curve. | This method results in "re-synchronization" of corticospinal action potentials at the level of the peripheral motor neuron and overcomes trial-to-trial variability in MEPs that is caused by phase cancellation and asynchronous firing of corticospinal motor neurons (*Rossini et al., 2015*). The main utility of TST is to examine corticospinal conduction deficits induced by demyelination (*Chen et al., 2008*; *Vucic et al., 2023*). | *Magistris et al. (1999)* |
| ***Silent periods*** | | | |
| Corticospinal silent period (CSP) | Also known as contralateral silent period (CSP). Quiescence in rectified EMG trace after MEP, when TMS is delivered during tonic contraction of target muscle (*Rossini et al., 2015*). CSP duration increases linearly with TMS stimulus intensity (stimulus-response curve) (*Rossini et al., 2015*). Reported as onset latency or duration of silent period. | Generated by spinal (recurrent inhibition, refractoriness of spinal motor neurons, post-synaptic inhibition) and intracortical inhibitory circuits (*Rossini et al., 2015*). The stimulus-response curve partly reflects gain and excitability characteristics of GABAergic inhibitory interneurons (*Rossini et al., 2015*; *Ziemann et al., 2015*). Short and long CSPs are mediated by $GABA_A$- and $GABA_B$-receptor activity, respectively (*Rossini et al., 2015*; *Ziemann et al., 2015*). The exact structural and functional mechanisms–including cortical *versus* spinal contributions–represent an area of intense scrutiny across the literature (*Hupfeld et al., 2020*; *Škarabot et al., 2019*; *Yacyshyn et al., 2016*). May index excitotoxicity (*Snow et al., 2019*). | *Tataroglu et al. (2003)* |
| Ipsilateral silent period (iSP) | Suppression of background rectified EMG trace following a suprathreshold TMS pulse, during tonic contraction of the homologous muscle ipsilateral to the target motor area (*Rossini et al., 2015*). Reported as onset latency, duration, depth, or transcallosal conduction time. | Reflects interhemispheric or transcallosal inhibition (*Wassermann et al., 1991*), the influence of one brain hemisphere over the other *via* projections across the *corpus* callosum or other commissural pathways (*Hupfeld et al., 2020*). Proxy of cortical glutamatergic and $GABA_B$ergic neuron activity (*Ferbert et al., 1992*; *Wassermann et al., 1991*). May index interhemispheric conduction loss or axonal damage (*Jung et al., 2006*; *Llufriu et al., 2012*; *Neva et al., 2016*; *Snow et al., 2019*). | *Jung et al. (2006)*, *Schmierer et al. (2000)*, *Schmierer et al. (2002)* |

**Note:**
AMPA, alpha-amino-3-hydroxy-5-methyl-4-isoxazolepropionic acid, ionotropic transmembrane glutamate receptor; EMG, electromyography; GABA, gamma-aminobutyric acid; $GABA_A$, ionotropic GABA receptor and ligand-gated ion (chloride, bicarbonate) channel; $GABA_B$, G-protein (*via* potassium channels) coupled metabotropic GABA receptor; MSO, maximum stimulator output; VGSC, voltage-gated sodium channel.

## Study screening

English, peer-reviewed journal articles of original studies were screened by a single author (NJS). Screening criteria were planned by the entire study team using the PICOS format (*Schardt et al., 2007*).

**Population**. Adults with MS; diagnosed using validated criteria; and sample size $n > 40$, to enhance sensitivity and specificity estimates (*Langer-Gould et al., 2006*; *Schaffler et al., 2011*).

**Intervention**. Observational research that reported sensitivity and/or specificity.

**Control**. Healthy controls, free of neurologic or other disease; persons with alternative diagnoses.

**Outcome**. Sensitivity and/or specificity of any upper and/or lower extremity TMS technique.

**Study**. Cross-sectional or case-control studies comparing MS to control participants; or cohort studies following participants from symptom onset to diagnosis.

## Data extraction

The approach to data extraction was planned by the entire study team and performed by a single author (NJS). The study team verified all transcribed data.

From study methods, a single author (NJS) transcribed study and participant characteristics, TMS methods, and criteria to determine abnormal TMS results relative to controls (*Chipchase et al., 2012*; *Langer-Gould et al., 2006*; *Schaffler et al., 2011*; *Snow et al., 2019*).

From study results, a single author (NJS) transcribed $2 \times 2$ contingency findings (*Adeniyi et al., 2016*; *Glas et al., 2003*; *McInnes et al., 2018*) (Table 2). Sensitivity was considered the percentage of participants with a diagnosis of MS, who exhibited abnormal TMS results (Sensitivity = True Positives ÷ (True Positives + False Negatives)). Specificity was considered the percentage of control participants (*i.e.*, without a diagnosis of MS), who exhibited normal TMS results (Specificity = True Negatives ÷ (False Positives + True Negatives)). All studies provided sufficient information to determine sensitivity. In cases where insufficient information was available to determine specificity, only sensitivity was reported. Sensitivity or specificity estimates below 50% indicated that the TMS outcome performed worse than chance at ruling out or in MS, respectively.

When both sensitivity and specificity outcomes were available, the diagnostic odds ratio (DOR) was estimated (*Glas et al., 2003*). DOR was calculated as DOR = (True Positives ÷ False Negatives) ÷ (False Positives ÷ True Negatives) (*Glas et al., 2003*). Any DOR above 1.0 was associated with increased diagnostic accuracy (*i.e.*, an increased odds that an abnormal TMS result was associated with diagnosis of MS) (*Glas et al., 2003*). DOR values were interpreted as trivial if < 1.68, small if 1.68–3.46, medium if 3.47–6.71, and large if > 6.71 (*Chen, Cohen & Chen, 2010*). When possible, 95% confidence intervals for sensitivity, specificity, and DOR were estimated using the methods outlined by *Glas et al. (2003)*.

**Table 2 Sample 2 × 2 contingency table.**

| | | MS diagnosis | | |
| --- | --- | --- | --- | --- |
| | | Positive | Negative | Total |
| TMS results | Abnormal | True positive (TP) | False positive (FP) | TP+FP |
| | Normal | False negative (FN) | True negative (TN) | FN+TN |
| | Total | TP+FN | FP+TN | FN+TN+FP+TP |

**Note:**
MS, multiple sclerosis; TMS, transcranial magnetic stimulation.

## Critical appraisal

The approach to critical appraisal was planned by the entire study team and completed by a single author (NJS). The study team verified all critical appraisal findings.

**Risk of bias.** Risk of bias was assessed using the QUADAS-2 tool (*Whiting et al., 2011*), which evaluates studies' reporting in domains of participant selection, the index test (TMS), the reference standard (MS diagnostic criteria), and participant flow and timing.

Participant selection questions evaluated reporting of the participant selection process and the level of detail used to describe participant samples:

i) Was a consecutive or random sample of patients enrolled?
ii) Was a case-control design avoided?
iii) Did the study avoid inappropriate exclusions?

Index test questions assessed reporting of TMS data collection, analysis, interpretation, and summarization:

i) Were the index test results interpreted without knowledge of the results of the reference standard (*i.e.*, was blinding employed)?
ii) If a threshold was used, was it pre-specified?

Reference standard questions examined reporting of MS diagnostic criteria:

i) Is the reference standard likely to correctly classify the target condition?
ii) Were the reference standard results interpreted without knowledge of the results of the index test (*i.e.*, was blinding employed)?

Flow and timing questions appraised reporting of participant exclusions and the timing between MS diagnosis and TMS testing:

i) Was there an appropriate interval between index test and reference standard?
ii) Did all patients receive a reference standard?
iii) Did all patients receive the same reference standard?
iv) Were all patients included in the analysis?

A single author (NJS) answered signaling questions as Yes/No/Unclear, to derive High/Low/Unclear risk of bias for each domain. Based on the risk of bias from each domain, an overall risk of bias rating was assigned to each study (*Sterne et al., 2019*).

**Biomarker validity.** To explore whether studies provided sufficient evidence to justify TMS techniques as biomarkers for MS diagnosis, *Bielekova & Martin*'s *(2004)* MS biomarker criteria were used. The *Bielekova & Martin (2004)* criteria classify biomarker studies according to MS-related pathophysiologic process, grade studies' methodologic quality, evaluate studies' clinical utility, and assess studies' clinical usefulness.

MS-specific pathophysiologic processes were classified as:

i) Biomarkers reflecting alteration of the immune system,
ii) Biomarkers of blood-brain barrier (BBB) disruption,
iii) Biomarkers of demyelination,
iv) Biomarkers of oxidative stress and excitotoxicity,
v) Biomarkers of axonal/neuronal damage,
vi) Biomarkers of gliosis, and/or
vii) Biomarkers of remyelination and repair.

Methodologic quality of studies was based on the following questions:

i) Are complete (raw) data provided?
ii) Was there an independent comparison to a reference standard or age- and sex-matched reference group?
iii) Was an appropriate spectrum of patients included (*e.g.*, clinical subtypes, sample size)?
iv) Were the methods used valid (*e.g.*, data collection, processing, and analysis)?
v) Was there a processing and/or work-up bias (*e.g.*, blinded processing and analysis)?

Clinical utility was evaluated according to the following criteria:

i) Biological rationale (*i.e.*, rational association with a pathogenic aspect of MS).
ii) Clinical relevance (*i.e.*, positioned in the causal chain of pathological events leading to a meaningful clinical endpoint).
iii) Practicality (*i.e.*, invasiveness of collection, need for serial analyses, reproducibility, ease, cost).
iv) Correlation with disease activity (*i.e.*, relationship with clinical [relapses, progression, disability scale] or neuroimaging [lesion numbers, atrophy] end points).
v) Correlation with disability/prognosis (*i.e.*, relationship with disability accumulation over time).
vi) Correlation with treatment effect (omitted).

Clinical usefulness was assessed against the following criteria:

i)  Sensitivity/specificity (*i.e.*, sensitivity/specificity relative to reference standard).
ii)  Reliability (*i.e.*, consistency of a measurement across time or raters, probability of false-positive or false-negative results).
iii)  Evaluation of a biomarker in epidemiological studies or natural history cohorts (*i.e.*, establishing a statistical relationship between the biomarker and clinical endpoint in cross-sectional or longitudinal studies).
iv)  Evaluation of a biomarker in proof-of-principle clinical trials (omitted).

A single author (NJS) applied ratings of Yes/No/Unclear to each criterion, to derive Yes/No/Unclear ratings for each domain. Based on ratings in each domain, an overall validity rating (Yes/No/Unclear) was assigned to each study (*Sterne et al., 2019*).

### Visual presentation of TMS outcomes

To visually compare the diagnostic accuracy of TMS techniques across studies, a single author (NJS) prepared a Forest plot of DOR point estimates and their 95% confidence intervals, organized by TMS outcome and study. DOR values were coded according to risk of bias rating. DOR values were interpreted as above (*Chen, Cohen & Chen, 2010*; *Glas et al., 2003*).

## RESULTS

See Fig. 2 for study selection (*Page et al., 2021*). The authors identified 964 records after duplicate removal. The authors reviewed 187 full texts after title and abstract screening and retained 17 articles for data extraction and critical appraisal.

### Study characteristics

Study design characteristics are summarized in Table S1. Only two studies prospectively followed participants from symptom presentation to diagnosis (*Beer, Rösler & Hess, 1995*; *Ravnborg et al., 1992*) and most used retrospective, case-control designs. No study followed participants with relapsing MS (RMS) to onset of secondary-progressive MS (SPMS); however, four studies cross-sectionally compared RMS to primary-progressive MS (PPMS) or SPMS (*Facchetti et al., 1997*; *Leocani et al., 2006*; *Schmierer et al., 2002*; *Tataroglu et al., 2003*). One study followed persons with active *vs* inactive RMS (*Caramia et al., 2004*). A total of 10 studies tested for subclinical lesions in participants without clinical motor findings (*Beer, Rösler & Hess, 1995*; *Cruz-Martínez et al., 2000*; *Hess et al., 1987*; *Jung et al., 2006*; *Kale et al., 2009*; *Kandler et al., 1991*; *Magistris et al., 1999*; *Mayr et al., 1991*; *Ravnborg et al., 1992*; *Tataroglu et al., 2003*). Two studies included comparison groups with diseases other than MS (*Beer, Rösler & Hess, 1995*; *Magistris et al., 1999*).

### Participant characteristics

MS participant characteristics are detailed in Table 3, while control group characteristics are summarized in Table S2. Across all studies, there were 1408 MS participants (median $n = 79$, range = 44–162) and 690 control participants (median $n = 34$, range = 10–155). Few studies matched MS and comparison groups for age or sex. Most studies used the

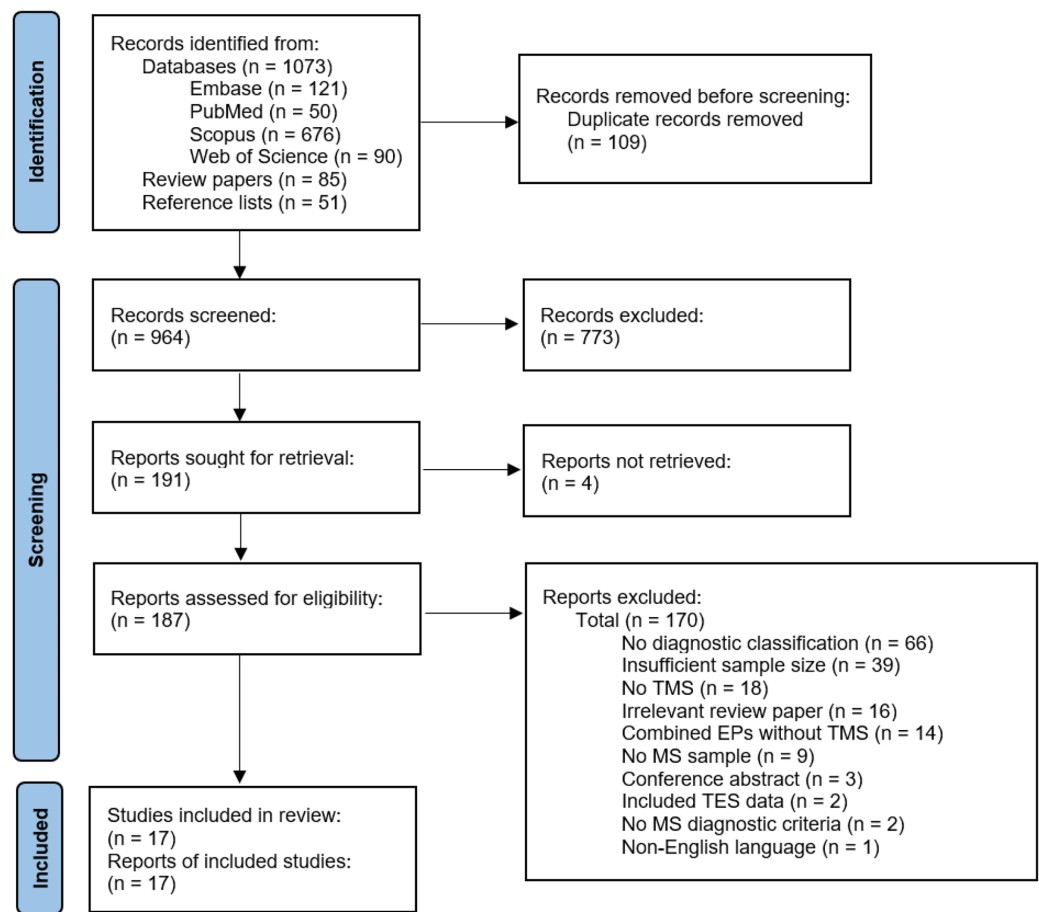

**Figure 2 Flow chart of study selection.** EP, evoked potential; MS, multiple sclerosis; TES, transcranial electrical stimulation; TMS, transcranial magnetic stimulation (*Page et al., 2021*).

Poser criteria to diagnose MS (*Poser et al., 1983*), whereas no study employed the 2017 McDonald criteria (*Thompson et al., 2018*). Average disease or symptom duration ranged from 1–10 years (median 4.6 years). Median disability score, measured using the Expanded Disability Status Scale (EDSS) (*Kurtzke, 1983*) was 2.5 (range 1.5–6).

## TMS findings

Table 4 and Fig. 3 summarize diagnostic accuracy findings of all TMS techniques studied. Table S3 highlights TMS methods and Table S4 provides an in-depth summary of TMS findings. Every study reported the sensitivity of the TMS techniques employed. Specificity and DOR could be gleaned from only eight studies (*Beer, Rösler & Hess, 1995*; *Cruz-Martínez et al., 2000*; *Hess et al., 1987*; *Magistris et al., 1999*; *Mayr et al., 1991*; *Ravnborg et al., 1992*; *Schmierer et al., 2000*; *Tataroglu et al., 2003*). Only central motor conduction time (CMCT) and MEP size (amplitude or area) were investigated by at least half of the studies reviewed. Therefore, only these techniques are discussed in detail below.

**CMCT.** Diagnostic accuracy findings for CMCT are summarized in Table 4 and Fig. 3. CMCT was the most frequently studied TMS technique, in 16 studies

**Table 3 Multiple sclerosis (MS) participant characteristics.**

| Study | Inclusion | Exclusion | Sample size | Diagnosis | MS Type | Females | Age | EDSS | Disease duration | Lesion location | Medication/drugs |
|---|---|---|---|---|---|---|---|---|---|---|---|
| *Beer, Rösler & Hess (1995)* | Suspected MS, admitted to inpatient neurology department | Pacemaker, seizure, intracranial neurosurgery, increased intracranial pressure, Any neurologic disorder, Age < 16 years | 142 | *Poser et al. (1983)* | NR | 80 | 37 (16–66) | NR | 2.9 (0–25) | Periventricular, Infratentorial | NR |
| *Caramia et al. (2004)* | Definite RMS, active or inactive, EDSS 0–3, ≥ 2 relapses | NR | 79 | *McDonald et al. (2001)* | 47 inactive RMS, 32 active RMS | 39 inactive RMS, 27 active RMS | Inactive RMS 37.8 (20–55), Active RMS 35.1 (18–52) | Inactive RMS 1 (0–2), Active RMS 2 (1–3) | Inactive RMS 4.5 (1–15), Active RMS 4.1 (1–11) | Centrum semiovale, subcortical | None |
| *Cruz-Martínez et al. (2000)* | Definite MS | Pacemaker, seizure, or intracranial neurosurgery | 50 | *Poser et al. (1983)* | NR | 31 | 31.9 (15–58) | NR | 4.2 (0–15) | 98% subcortical, internal capsule, mesencephalon, pons, 86% sensory pathways, 67% cervical spinal cord, 62% cerebellum, 44% brainstem, 40% optic radiations | NR |
| *Facchetti et al. (1997)* | Definite MS, outpatients at neurology clinic | PPMS, probable MS, or possible MS | 53 | *Poser et al. (1983)* | 40 RMS, 13 SPMS | 30 RMS, 10 SPMS | RMS 36 ± 10, SPMS 40 ± 7.4 | RMS 2 ± 1, SPMS 5 ± 1 | RMS 8.1 ± 7.5, SPMS 11.8 ± 5.8 | Subcortical | NR |
| *Hess et al. (1987)* | Definite, probable, or possible MS, referred for neurophysiological investigation or admitted to inpatient neurology department | Pacemaker, seizure, intracranial neurosurgery | 83 | *Poser et al. (1983)* | NR | NR | 41.4 (18–68) | NR | 8.3 (0–33) | NR | NR |
| *Jung et al. (2006)* | Definite or possible MS, inactive disease for ≥6 weeks, EDSS ≤ 4 | NR | 49 | *McDonald et al. (2001)* | 49 inactive RMS | 27 | 35.2 (23–54) | 1.5 (0–4) | 1 ± 1.5 | 79% *corpus* callosum, supratentorial, brainstem, cerebellum | Interferon Beta, Glatiramer Acetate No steroids |
| *Kale et al. (2009)* | Definite MS, inactive disease, outpatients at neurology clinic | Pacemaker, seizure, intracranial neurosurgery, head trauma, other metallic implant, Active disease in ≤8 weeks | 131 | *Poser et al. (1983)* | 73 inactive RMS, 43 inactive SPMS, 15 inactive PPMS | 111 | 36 ± 8 | 66 participants 0–2, 33 participants 2–4, 32 participants >4 | NR | NR | No steroids |

| Study | Inclusion | Exclusion | Sample size | Diagnosis | MS Type | Females | Age | EDSS | Disease duration | Lesion location | Medication/ drugs |
|---|---|---|---|---|---|---|---|---|---|---|---|
| Kale, Agaoglu & Tanik (2010) | Definite MS, inactive RMS or SPMS outpatients at neurology clinic | Pacemaker, seizure, intracranial neurosurgery, head trauma, other metallic implant, Active disease in ≤8 weeks | 79 | McDonald et al. (2001) | 60 inactive RMS, 19 inactive SPMS | 51 | 35.3 ± 7.6 | 41 participants 0–2, 19 participants 2–4, 19 participants > 4 | 32 participants <5, 31 participants 5–10, 16 participants >10 | 39% corpus callosum atrophy | No steroids |
| Kandler et al. (1991) | Definite, probable, possible, or suspected MS | Pacemaker, seizure, intracranial neurosurgery | 162 | McDonald & Halliday (1977) | NR | 112 | 38 (16–75) | NR | NR | NR | NR |
| Leocani et al. (2006) | Definite MS, inactive disease, outpatients at neurology clinic, complete neurologic exam and evoked potentials <3 weeks apart | Active disease in ≤2 months | 84 | Poser et al. (1983) | 43 inactive RMS, 28 inactive SPMS, 13 inactive PPMS | 28 inactive RMS, 18 inactive SPMS, 5 inactive PPMS | Inactive RMS 33.7 ± 9.4, Inactive SPMS 41 ± 9.5, Inactive PPMS 43.8 ± 6.9 | Inactive RMS 3 (1–6), Inactive SPMS 5 (2.5–8), Inactive PPMS 5.5 (2–6.5) | Inactive RMS 7.1 ± 5.9, Inactive SPMS 9.4 ± 7.1, Inactive PPMS 4.5 ± 5.5 | NR | No steroids |
| Magistris et al. (1999) | Definite, probable, or suspected MS, referred for neurophysiological investigation | NR | 116 | Poser et al. (1983) | NR | NR | 39 (17–76) | NR | NR | NR | NR |
| Mayr et al. (1991) | Definite MS | NR | 44 | Poser et al. (1983) | NR | 31 | 37.8 | NR | 7.9 | NR | NR |
| Pisa et al. (2020) | Definite PrMS, diagnosed ≥12 months prior, admitted for neurorehabilitation, referred for neuromodulation/ neurorehabilitation, inactive disease for ≥6 months, EDSS 4–6.5, Pyramidal functional systems score ≥3, cerebellar score ≤2, and cerebral score <2 | Pacemaker, seizure, intracranial neurosurgery, head trauma, other metallic implant, stroke, pregnancy, Comorbidity affecting ambulation, Recent botulinum toxin treatment | 50 | Revised McDonald (2010), Polman et al. (2011) | 32 inactive SPMS, 18 inactive PPMS | 27 | 49.4 ± 7.5 | 5.8 ± 0.7 | ≥1 | NR | No steroids |
| Ravnborg et al. (1992) | Suspected MS, admitted to inpatient neurology department | Any neurologic disorder | 68 | Poser et al. (1983) | 40 diagnosed as MS, 28 diagnosed as no MS | 38 | 40 (18–63) | NR | 1 (0–20) | Corticospinal tract, periventricular, subcortical, cerebellum | NR |

(Continued)

Snow et al. (2024), *PeerJ*, DOI 10.7717/peerj.17155

## Table 3 (continued)

| Study | Inclusion | Exclusion | Sample size | Diagnosis | MS Type | Females | Age | EDSS | Disease duration | Lesion location | Medication/drugs |
|---|---|---|---|---|---|---|---|---|---|---|---|
| *Schmierer et al. (2000)* | Definite MS | NR | 50 | *Poser et al. (1983)* | 50 RMS | 32 | 33 (16–52) | 2 (0–4.5) | 2.4 (1–6) | 88% middle/posterior *corpus* callosum, 69% pericallosal, 56% anterior *corpus* callosum | NR |
| *Schmierer et al. (2002)* | Definite MS, no active disease for ≥3 months | NR | 118 | *Poser et al. (1983)* | 96 inactive RMS, 19 inactive PPMS, 3 inactive SPMS | 76 | 37 (16–65) | 2.9 (0–6.5) | 4.9 (1–21) | NR | No steroids |
| *Tataroglu et al. (2003)* | Definite MS, inactive disease, outpatients at neurology clinic | Any neurologic or systemic disease | 50 | *McDonald et al. (2001)* | 37 RMS, 21 PrMS, | 38 | RMS 28.6 (17–49), PrMS 42.2 (26-56) | RMS 1.7 ± 1.2, PrMS 4.5 ± 1.9 | 6.7 (1–22) | 82% periventricular/*corpus* callosum, 12% brainstem and periventricular, 7% brainstem and cervical spinal cord | No steroids |

**Notes:**

Age and disease duration are reported in years. Continuous data are expressed as median (range) or mean ± standard deviation. DMT, disease modifying therapy; EDSS, Expanded Disability Status Scale; MS, multiple sclerosis; NR, not reported; PPMS, primary progressive multiple sclerosis; PrMS, progressive MS; SPMS, secondary progressive MS.

**Table 4 Transcranial magnetic stimulation (TMS) results.**

| Study | Sensitivity (95% CI) | Specificity (95% CI) | Diagnostic odds ratio (95% CI) | Associations with disease-related outcomes |
|---|---|---|---|---|
| **Resting motor threshold (RMT), one study (6%)** | | | | |
| Cruz-Martínez et al. (2000) | Upper extremity: 39% <br> Lower extremity: 43% | Upper extremity: 100% [98–100%] <br> Lower extremity: 100% [98–100%] | Upper extremity: 23.30 [13.66–39.75] <br> Lower extremity: 27.75 [13.22–58.23] | RMT was correlated with EDSS ($p < 0.02$), ataxia ($p < 0.04$), and central motor pathway MRI lesions ($p < 0.05$). Magnitude not reported. |
| Schmierer et al. (2002) | RMS (upper + lower extremity): 18% <br> PPMS (upper + lower extremity): 10% | NR | NR | RMT was not significantly correlated with EDSS. |
| **Motor evoked potential (MEP), 10 studies (59%)** | | | | |
| *MEP size (amplitude, area), nine studies (53%)* | | | | |
| Cruz-Martínez et al. (2000) | Upper extremity: 24% <br> Lower extremity: 29% | Upper extremity: 100% [98–100%] <br> Lower extremity: 100% [98–100%] | Upper extremity: 11.60 [6.47–20.81] <br> Lower extremity: 14.80 [6.65–32.92] | MEP amplitude was correlated with EDSS ($p < 0.03$), ataxia ($p < 0.007$), and MRI lesions in the pons ($p < 0.009$) and cervical cord ($p < 0.03$). Magnitude not reported. |
| Hess et al. (1987) | Upper extremity: 47% | Upper extremity: 100% [94–100%] | Upper extremity: 27.48 [15.81–47.78] | NR |
| Kale et al. (2009) | Upper extremity: 83% [82–84%] | NR | NR | MEP amplitude was correlated with EDSS ($p < 0.001$). Magnitude not reported. |
| Kale, Agaoglu & Tanik (2010) | Upper extremity: 85% (83–87%) | NR | NR | MEP amplitude was correlated with EDSS ($p < 0.05$) and *corpus* callosum atrophy ($p$ not reported). Magnitude not reported. |
| Kandler et al. (1991) | Upper extremity: 9% <br> Lower extremity: 25% | NR | NR | MEP amplitude was correlated with pyramidal dysfunction (hyperreflexia, weakness, spasticity, plantar reflex) ($p$ not reported). Magnitude not reported. |
| Mayr et al. (1991) | Upper extremity: 11% <br> Lower extremity: 28% | Upper extremity: 99% [97–100%] <br> Lower extremity: 100% [98–100%] | Upper extremity: 2.63 [0.67–10.34] * <br> Lower extremity: 7.69 [2.31–25.61] | MEP amplitude was not significantly correlated with pyramidal dysfunction (hyperreflexia, weakness, spasticity, plantar reflex). |
| Ravnborg et al. (1992) | Upper + lower extremity: 50% [38–63%] | Upper + lower extremity: 86% [81–100%] | Upper + lower extremity: 6.00 [1.76–20.46] | MEP amplitude was correlated with MRI lesion number (McNemar's = 0.85, $p$ not reported) but not pyramidal dysfunction (hyperreflexia, weakness, spasticity, plantar reflex). |
| Schmierer et al. (2000) | Upper extremity: 34% <br> Lower extremity: 6% | Upper extremity: 100% [92–100%] <br> Lower extremity: 100% [92–100%] | Upper extremity: 12.36 [6.11–25.00] <br> Lower extremity: 1.53 [0.45–5.24] * | MEP amplitude was not significantly correlated with MRI lesion location or burden. |
| Tataroglu et al. (2003) | Upper + lower extremity: 66% [64–68%] | Upper + lower extremity: 94% [88–100%] | Upper + lower extremity: 27.55 [5.95–127.46] | MEP amplitude was not significantly correlated with EDSS. |

| Study | Sensitivity (95% CI) | Specificity (95% CI) | Diagnostic odds ratio (95% CI) | Associations with disease-related outcomes |
|---|---|---|---|---|
| *MEP latency, four studies (24%)* | | | | |
| *Kale et al. (2009)* | Upper extremity: 52% [52–52%] | NR | NR | MEP latency was correlated with EDSS ($p < 0.001$). Magnitude not reported. |
| *Kale, Agaoglu & Tanik (2010)* | Upper extremity: 43% | NR | NR | MEP latency was correlated with *corpus* callosum atrophy ($p$ not reported) but not EDSS. Magnitude not reported. |
| *Pisa et al. (2020)* | Upper extremity: 82% [79–85%] Lower extremity: 98% [94–100%] | NR | NR | Upper extremity MEP latency was correlated with EDSS (Rho = 0.296, $p < 0.05$) and walking performance (Rho = 0.6, $p < 0.0001$). Lower extremity MEP latency not reported. |
| *Tataroglu et al. (2003)* | Upper + lower extremity: 69% [67–71%] | Upper + lower extremity: 80% [75–85%] | Upper + lower extremity: 9.26 [3.24–26.47] | MEP latency was not significantly correlated with EDSS. |
| ***Central motor conduction time (CMCT), 16 studies (94%)*** | | | | |
| *Beer, Rösler & Hess (1995)* | Upper + lower extremity: 68% [67–69%] | Upper + lower extremity: 77% [74–80%] | Upper + lower extremity: 6.83 [3.19–14.62] | NR |
| *Caramia et al. (2004)* | Upper extremity: 16% | NR | NR | NR |
| *Cruz-Martínez et al. (2000)* | Upper extremity: 61% [60–62%] Lower extremity: 51% [50–52%] | Upper extremity: 100% [98–100%] Lower extremity: 100% [98–100%] | Upper extremity: 58.76 [34.45–100.24] Lower extremity: 39.17 [18.78–81.70] | CMCT was correlated with EDSS ($p < 0.01$), pyramidal dysfunction (hyperreflexia, weakness, spasticity, plantar reflex) ($p < 0.02$), ataxia ($p < 0.02$), and MRI lesions in the pons ($p < 0.03$) and central motor pathway ($p < 0.04$). Magnitude not reported. |
| *Facchetti et al. (1997)* | RMS (upper extremity): 30% SPMS (upper extremity): 100% [85–100%] RMS (lower extremity): 43% SPMS (lower extremity): 100% [85–100%] | NR | NR | CMCT was not significantly correlated with EDSS or number or area of MRI lesions. |
| *Hess et al. (1987)* | Upper extremity: 72% [70–74%] | Upper extremity: 100% [94–100%] | Upper extremity: 80.87 [44.71–146.26] | CMCT was correlated with hyperreflexia ($p < 0.001$), weakness ($p < 0.05$), and ataxia ($p < 0.05$), but not impaired fine movements or sensory deficits. Magnitude not reported. |
| *Jung et al. (2006)* | Upper extremity: 25% Lower extremity: 69% [68–70%] | NR | NR | Upper extremity, but not lower extremity, CMCT was correlated with pyramidal dysfunction (hyperreflexia, weakness, spasticity, plantar reflex) ($p < 0.005$), but not *corpus* callosum atrophy or MRI lesion volume or number. Magnitude not reported. |

| Study | Sensitivity (95% CI) | Specificity (95% CI) | Diagnostic odds ratio (95% CI) | Associations with disease-related outcomes |
|---|---|---|---|---|
| *Kale et al. (2009)* | Upper extremity: 49% | NR | NR | CMCT was correlated with EDSS ($p < 0.001$). Magnitude not reported. |
| *Kale, Agaoglu & Tanik (2010)* | Upper extremity: 41% | NR | NR | CMCT was correlated with *corpus* callosum atrophy ($p$ not reported) but not EDSS. Magnitude not reported. |
| *Kandler et al. (1991)* | Upper extremity: 43% Lower extremity: 67% [66–68%] | NR | NR | NR |
| *Leocani et al. (2006)* | RMS (upper extremity): 56% [54–58%] SPMS (upper extremity): 93% [87–99%] PPMS (upper extremity): 85% [72–98%] RMS (lower extremity): 61% RMS [59–63%] SPMS (lower extremity): 96% SPMS [89–100%] PPMS (lower extremity): 92% PPMS [78–100%] | NR | NR | CMCT was correlated with EDSS (Rho = 0.6, $p < 0.001$). |
| *Magistris et al. (1999)* | Upper extremity: 27% | Upper extremity: 58% [58–58%] | Upper extremity: 0.52 [0.35–0.76]‡ | CMCT was not significantly correlated with weakness. |
| *Mayr et al. (1991)* | Upper extremity: 71% [68–74%] Lower extremity: 61% [59–63%] | Upper extremity: 99% [97–100%] Lower extremity: 100% ]98–100%] | Upper extremity: 202.69 [102.56–400.59] Lower extremity: 135.00 [71.00–256.69] | CMCT was correlated with pyramidal dysfunction (hyperreflexia, weakness, spasticity, plantar reflex) ($p$ not reported). Magnitude not reported. |
| *Ravnborg et al. (1992)* | Upper + lower extremity: 83% [73–93%] | Upper + lower extremity: 75% [61–89%] | Upper + lower extremity: 14.14 [4.34–46.11] | CMCT was correlated with MRI lesion number (McNemar's = 0.85, $p$ not reported) but not pyramidal dysfunction (hyperreflexia, weakness, spasticity, plantar reflex). |
| *Schmierer et al. (2000)* | Upper extremity: 14% Lower extremity: 48% | Upper extremity: 100% [92–100%] Lower extremity: 100% [92–100%] | Upper extremity: 3.91 [1.61–9.52] Lower extremity: 22.15 [11.23–43.69] | CMCT was not significantly correlated with MRI lesion burden or location. |
| *Schmierer et al. (2002)* | RMS (upper extremity): 32% PPMS (upper extremity): 37% RMS (lower extremity): 63% [60–66%] PPMS (lower extremity): 58% [56–60%] | NR | NR | Upper and lower extremity CMCT was correlated with EDSS ($r = 0.4$–0.5, $p < 0.01$). |
| *Tataroglu et al. (2003)* | Upper + lower extremity: 76% [74–78%] | Upper + lower extremity: 87% [82–92%] | Upper + lower extremity: 21.21 [6.32–71.14] | CMCT was not significantly correlated with EDSS. |

(Continued)
| Table 4 (continued) | | | | |
|---|---|---|---|---|
| **Study** | **Sensitivity (95% CI)** | **Specificity (95% CI)** | **Diagnostic odds ratio (95% CI)** | **Associations with disease-related outcomes** |
| *Triple stimulation technique (TST), one study (6%)* | | | | |
| *Magistris et al. (1999)* | Upper extremity: 48% | Upper extremity: 60% [60–60%] | Upper extremity: 0.60 [0.42–0.86]‡ | TST amplitude ratio was correlated with weakness (*p* < 0.0001). Magnitude not reported. |
| *Corticospinal silent period (CSP), one study (6%)* | | | | |
| *Tataroglu et al. (2003)* | Upper + lower extremity: 69% [67–71%] | Upper + lower extremity: 70% [66–74%] | Upper + lower extremity: 5.43 [2.09–14.10] | CSP duration was correlated with ataxia (*r* = 0.3, *p* < 0.001) but not EDSS. |
| *Ipsilateral silent period (iSP), three studies (16%)* | | | | |
| *iSP latency, three studies (16%)* | | | | |
| *Jung et al. (2006)* | Upper extremity: 4% | NR | NR | iSP latency was not significantly correlated with pyramidal dysfunction (hyperreflexia, weakness, spasticity, plantar reflex), *corpus* callosum atrophy, or MRI lesion volume or number. |
| *Schmierer et al. (2000)* | Upper extremity: 18% | Upper extremity: 100% [92–100%] | Upper extremity: 5.27 [2.32–11.98] | iSP latency was not significantly correlated with MRI lesion burden or location. |
| *Schmierer et al. (2002)* | Upper extremity RMS: 16% Upper extremity PPMS: 34% | NR | NR | iSP latency was correlated with EDSS in PPMS (*r* = 0.4, *p* < 0.01) but not RMS. |
| *iSP duration, three studies (18%)* | | | | |
| *Jung et al. (2006)* | Upper extremity: 22% | NR | NR | iSP duration was not significantly correlated with pyramidal dysfunction (hyperreflexia, weakness, spasticity, plantar reflex), *corpus* callosum atrophy, or MRI lesion volume or number. |
| *Schmierer et al. (2000)* | Upper extremity: 72% [69–75%] | Upper extremity: 100% [92–100%] | Upper extremity: 61.71 [29.70–128.22] | iSP duration was with MRI lesion burden (*r* = 0.4, *p* < 0.01) but not MRI lesion location. |
| *Schmierer et al. (2002)* | Upper extremity RMS: 16% Upper extremity PPMS: 34% | NR | NR | iSP duration was not significantly correlated with EDSS. |
| *iSP depth, one study (6%)* | | | | |
| *Jung et al. (2006)* | Upper extremity: 6% | NR | NR | iSP depth was not significantly correlated with pyramidal dysfunction (hyperreflexia, weakness, spasticity, plantar reflex), *corpus* callosum atrophy, or MRI lesion volume or number. |
| *Transcallosal conduction time (TCT), three studies (18%)* | | | | |
| *Jung et al. (2006)* | Upper extremity: 6% | NR | NR | TCT was not significantly correlated with pyramidal dysfunction (hyperreflexia, weakness, spasticity, plantar reflex), *corpus* callosum atrophy, or MRI lesion volume or number. |
| *Schmierer et al. (2000)* | Upper extremity: 4% | Upper extremity: 100% [92–100%] | Upper extremity: 1.00 [0.23–4.34]* | TCT was not significantly correlated with MRI lesion burden or location. |

| Study | Sensitivity (95% CI) | Specificity (95% CI) | Diagnostic odds ratio (95% CI) | Associations with disease-related outcomes |
|---|---|---|---|---|
| *Schmierer et al. (2002)* | Upper extremity RMS: 13%<br>Upper extremity PPMS: 24% | NR | NR | TCT was not significantly correlated with EDSS. |

**Notes:**
EDSS, Expanded Disability Status Scale; MRI, magnetic resonance imaging; NR, not reported; PPMS, primary progressive multiple sclerosis; RMS, relapsing multiple sclerosis; SPMS, secondary progressive multiple sclerosis; TN, true negative; TP, true positive. *, 95% CI of diagnostic odds ratio (DOR) crossed zero, suggesting no change in odds of MS. ‡, DOR < 1 indicated decreased odds of MS.

(*Beer, Rösler & Hess, 1995*; *Caramia et al., 2004*; *Cruz-Martínez et al., 2000*; *Facchetti et al., 1997*; *Hess et al., 1987*; *Jung et al., 2006*; *Kale et al., 2009*; *Kale, Agaoglu & Tanik, 2010*; *Kandler et al., 1991*; *Leocani et al., 2006*; *Magistris et al., 1999*; *Mayr et al., 1991*; *Ravnborg et al., 1992*; *Schmierer et al., 2002, 2000*; *Tataroglu et al., 2003*). For RMS, sensitivity was greatest when upper and lower limbs were combined (median 76%, range 68–83%). Lower extremity CMCT was most specific for RMS (median 100%, range 100–100%), and was associated with the greatest odds of diagnosis (median DOR 39.17, range 22.15–135.00).

CMCT had poor sensitivity for subclinical lesions (median 19%, range 6–59%), but was up to 96% specific. CMCT tended to be more sensitive for both PPMS (median 72%, range 37–92%) and SPMS (median 96%, range 93–100%) than RMS (median 50%, range 30–63%) in head-to-head comparisons. In participants with RMS, CMCT had negligible sensitivity for detecting the onset of new disease activity (*Caramia et al., 2004*), but normalization of CMCT was 75% sensitive for the recovery from active to inactive disease (*Cruz-Martínez et al., 2000*).

**MEP size.** Diagnostic accuracy findings for MEPs are summarized in Table 4 and Fig. 3. MEP size (amplitude or area) was the second-most frequently studied TMS technique, in nine studies (*Cruz-Martínez et al., 2000*; *Hess et al., 1987*; *Kale et al., 2009*; *Kale, Agaoglu & Tanik, 2010*; *Kandler et al., 1991*; *Mayr et al., 1991*; *Ravnborg et al., 1992*; *Schmierer et al., 2000*; *Tataroglu et al., 2003*). Like CMCT, combined findings from upper and lower extremities yielded the highest sensitivity for RMS (median 58%, range 50–66%). When examined separately, the upper and lower extremities were most specific for RMS (median 100%, range 100–100%), compared to combined upper and lower extremities (median 90%, range 86–94%). However, DOR was greatest for combined upper and lower extremities (median 16.78, range 6.00–27.55). MEP size was poorly sensitive for subclinical lesions (median 29%, range 15–67%) and was not examined in relation to PPMS or SPMS, nor in active *vs* inactive RMS.

**Associations between TMS and disease-related outcomes.** Table 4 summarizes associations between TMS techniques and disease-related outcomes. Both CMCT and MEP were correlated with cerebellar function (*Cruz-Martínez et al., 2000*; *Hess et al., 1987*; *Leocani et al., 2006*; *Tataroglu et al., 2003*). CMCT and MEP size were associated with pyramidal function in some studies (*Cruz-Martínez et al., 2000*; *Hess et al., 1987*; *Jung et al., 2006*; *Kandler et al., 1991*; *Leocani et al., 2006*; *Magistris et al., 1999*;

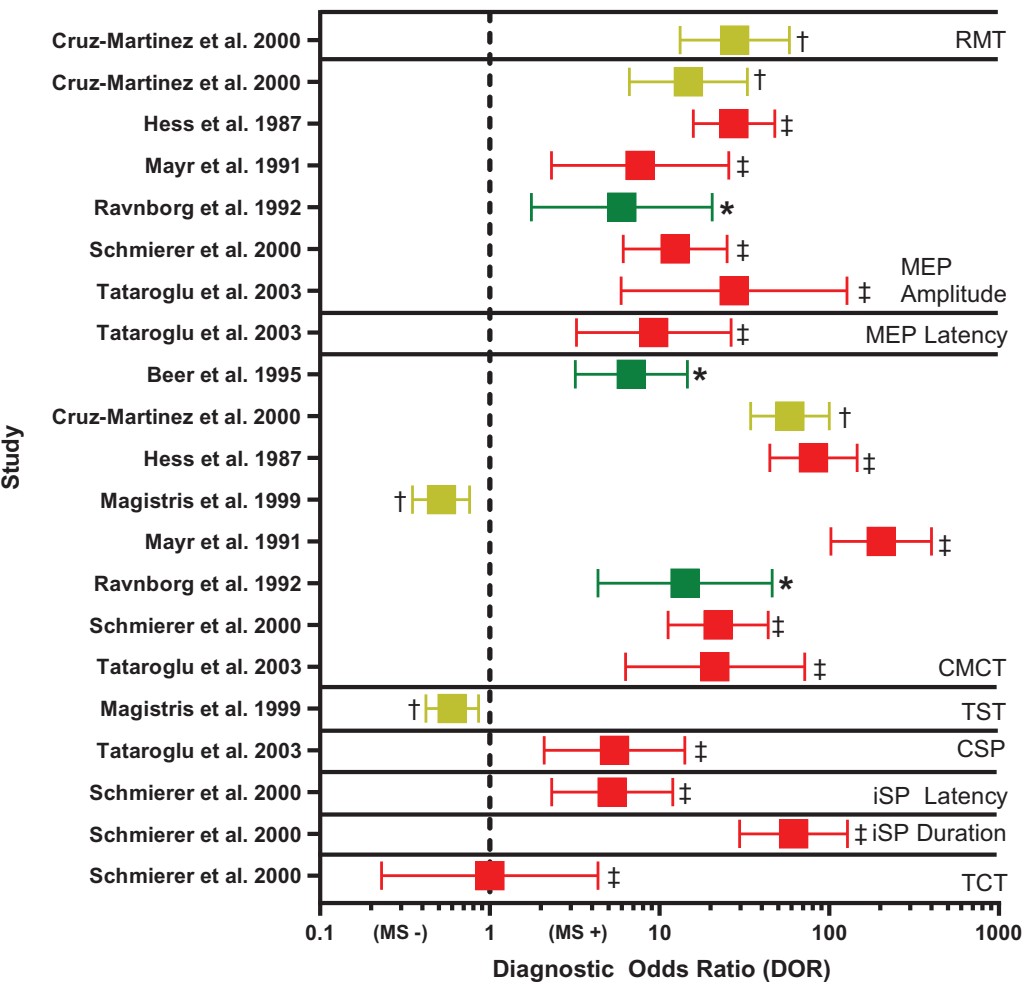

**Figure 3 Forest plot of diagnostic odds ratio (DOR) for transcranial magnetic stimulation (TMS)-based outcomes.** Demonstrates DOR point estimates and 95% confidence intervals studies, coded by risk of bias (* or green = Low, † or yellow = Unclear, ‡ or red = High). CMCT, central motor conduction time; CSP, corticospinal silent period; iSP, ipsilateral silent period; MEP, motor evoked potential; MS −, negative diagnosis of multiple sclerosis; MS +, positive diagnosis of multiple sclerosis; RMT, resting motor threshold; TCT, transcallosal conduction time; TST, triple stimulation technique. Note, for studies reporting multiple DORs, only the highest value is reported (*Cruz-Martínez et al., 2000*; *Hess et al., 1987*; *Mayr et al., 1991*; *Ravnborg et al., 1992*; *Schmierer et al., 2000*; *Tataroglu et al., 2003*; *Beer, Rösler & Hess, 1995*; *Magistris et al., 1999*).

*Mayr et al., 1991*), but not others (*Facchetti et al., 1997*; *Mayr et al., 1991*; *Ravnborg et al., 1992*). TMS outcomes were not related to other functional systems (*i.e.*, brainstem, sensory, visual, cerebral, bowel-bladder) (*Hess et al., 1987*; *Tataroglu et al., 2003*). EDSS was significantly associated with both CMCT and MEP size (*Cruz-Martínez et al., 2000*; *Facchetti et al., 1997*; *Kale et al., 2009*; *Kale, Agaoglu & Tanik, 2010*; *Leocani et al., 2006*; *Pisa et al., 2020*; *Schmierer et al., 2002*). Lastly, CMCT and MEP size were modestly associated with *corpus* callosum atrophy, total lesion burden, and corticospinal tract lesions some studies (*Cruz-Martínez et al., 2000*; *Kale, Agaoglu & Tanik, 2010*; *Ravnborg et al., 1992*), but not others (*Facchetti et al., 1997*; *Schmierer et al., 2000*).

### Risk of bias

See Table 5 for a condensed summary of overall risk of bias findings and Table S5 for more detailed results. Only two studies had low risk of bias (*Beer, Rösler & Hess, 1995*; *Ravnborg et al., 1992*), whereas the remainder had either unclear or high risk of bias. The principal source of bias was participant selection, resulting in high risk of bias in 12 studies (*Facchetti et al., 1997*; *Hess et al., 1987*; *Jung et al., 2006*; *Kale et al., 2009*; *Kale, Agaoglu & Tanik, 2010*; *Kandler et al., 1991*; *Leocani et al., 2006*; *Mayr et al., 1991*; *Pisa et al., 2020*; *Schmierer et al., 2002, 2000*; *Tataroglu et al., 2003*), 10 of which did not avoid a case-control design (*Facchetti et al., 1997*; *Hess et al., 1987*; *Jung et al., 2006*; *Kale et al., 2009*; *Kale, Agaoglu & Tanik, 2010*; *Kandler et al., 1991*; *Mayr et al., 1991*; *Pisa et al., 2020*; *Schmierer et al., 2000*; *Tataroglu et al., 2003*).

### Biomarker assessment

See Table 6 for a condensed summary of biomarker assessment findings and Table S6 for more detailed results. No study demonstrated sufficient validity for TMS biomarker use (*Bielekova & Martin, 2004*). Specifically, no study supported TMS use in epidemiologic research. Nearly all studies both failed to provide raw or participant-level data, and failed to either establish or adequately report sensitivity and/or specificity (*Beer, Rösler & Hess, 1995*; *Caramia et al., 2004*; *Cruz-Martínez et al., 2000*; *Facchetti et al., 1997*; *Hess et al., 1987*; *Jung et al., 2006*; *Kale et al., 2009*; *Kale, Agaoglu & Tanik, 2010*; *Kandler et al., 1991*; *Leocani et al., 2006*; *Magistris et al., 1999*; *Mayr et al., 1991*; *Pisa et al., 2020*; *Ravnborg et al., 1992*; *Schmierer et al., 2002, 2000*; *Tataroglu et al., 2003*). Twelve studies did not address the reliability of TMS techniques (*Beer, Rösler & Hess, 1995*; *Cruz-Martínez et al., 2000*; *Facchetti et al., 1997*; *Jung et al., 2006*; *Kale et al., 2009*; *Kale, Agaoglu & Tanik, 2010*; *Kandler et al., 1991*; *Mayr et al., 1991*; *Pisa et al., 2020*; *Ravnborg et al., 1992*; *Schmierer et al., 2000*; *Tataroglu et al., 2003*). However, despite the above shortcomings, nearly all studies justified disease process-specific biological rationale for TMS to investigate MS, namely the ability to characterize demyelination and/or axonal damage.

## DISCUSSION

This review aimed to summarize the diagnostic accuracy and validity of TMS techniques to aid in the diagnosis of MS. Across all TMS techniques studied, there was modest sensitivity for MS at best. Of the few studies that evaluated specificity, only CMCT and MEP size (amplitude or area) were represented in enough studies to comment on overall diagnostic performance. Most studies had a high risk of bias and did not demonstrate validity for TMS biomarker use.

### Diagnostic performance of TMS techniques

Recent reviews have discussed the role of TMS as a biomarker in MS (*Alsharidah et al., 2022*; *Simpson & Macdonell, 2015*; *Snow et al., 2019*; *Ziemann et al., 2011*). While there is optimism for using TMS to diagnose, monitor natural history, or assess treatment response in MS (*Alsharidah et al., 2022*), there is also hesitancy towards widespread clinical use of TMS due to lack of sufficient evidence and high risk of bias (*Simpson & Macdonell, 2015*;

**Table 5 Risk of bias assessment.**

| Study | Patient selection | Index test | Reference standard | Flow and timing | Risk of bias |
|---|---|---|---|---|---|
| Beer, Rösler & Hess (1995) | L | L | L | L | L |
| Caramia et al. (2004) | U | H | H | H | H |
| Cruz-Martínez et al. (2000) | U | L | L | U | U |
| Facchetti et al. (1997) | H | L | L | L | H |
| Hess et al. (1987) | H | H | H | H | H |
| Jung et al. (2006) | H | U | U | L | H |
| Kale et al. (2009) | H | U | U | L | H |
| Kale, Agaoglu & Tanik (2010) | H | U | U | L | H |
| Kandler et al. (1991) | H | U | U | H | H |
| Leocani et al. (2006) | H | L | L | L | H |
| Magistris et al. (1999) | L | U | U | L | U |
| Mayr et al. (1991) | H | U | U | L | H |
| Pisa et al. (2020) | H | U | U | L | H |
| Ravnborg et al. (1992) | L | L | L | L | L |
| Schmierer et al. (2000) | H | U | U | L | H |
| Schmierer et al. (2002) | H | U | U | H | H |
| Tataroglu et al. (2003) | H | U | U | U | H |

**Note:**

H, high; L, low; U, unclear. See ref: (Whiting et al., 2011).

Snow et al., 2019). In the current review, studies failed to justify the validity of TMS techniques as biomarkers for MS diagnosis; however, some outcomes could help characterize corticospinal conduction loss throughout the disease course.

TMS elicits corticospinal motor responses partly by stimulating axon terminals or axonal bends in superficial presynaptic layer II/III/V myelinated neurons at the gyral crown of the precentral gyrus (Siebner et al., 2022). Given MS is characterized by attacks of inflammatory demyelination and axonal damage (in the context of ongoing axonal degeneration) (Pachner, 2021; Reich, Lucchinetti & Calabresi, 2018), these processes can intuitively be characterized using TMS measures of CNS conduction (Vucic et al., 2023; Ziemann et al., 2011). For example, CMCT, MEPs, and triple stimulation technique (TST) could theoretically serve this role (Alsharidah et al., 2022; Simpson & Macdonell, 2015; Snow et al., 2019; Vucic et al., 2023; Ziemann et al., 2011). In the present review, only CMCT and MEP size had sufficient evidence to evaluate their diagnostic performance.

Only CMCT was sensitive for MS (median 75%), when combining findings from upper plus lower extremities. Both CMCT and MEP amplitude had high specificity for MS (median 100%). DOR was greatest and most consistent for CMCT of the lower extremities (median DOR 25.15). Both techniques were generally correlated with pyramidal function, cerebellar function, and EDSS; however, their associations with MRI findings (lesions, atrophy) were inconsistent. Both techniques had poor sensitivity for subclinical lesions, whereas CMCT was 96% specific for subclinical disease. CMCT was 75% sensitive for recovery from active disease but had negligible sensitivity for new disease activity. Lastly, CMCT was highly sensitive for SPMS and PPMS.

**Table 6 Biomarker assessment.**

| Study | Process-specific classification | Methodologic quality | Clinical utility | Clinical usefulness | Biomarker validity |
|---|---|---|---|---|---|
| *Beer, Rösler & Hess (1995)* | Axonal damage | N | U | N | N |
| *Caramia et al. (2004)* | Demyelination, axonal damage | N | N | N | N |
| *Cruz-Martínez et al. (2000)* | Demyelination, axonal damage | N | Y | N | N |
| *Facchetti et al. (1997)* | Demyelination, axonal damage | N | U | U | N |
| *Hess et al. (1987)* | Demyelination, axonal damage | N | U | N | N |
| *Jung et al. (2006)* | Demyelination, axonal damage | N | N | N | N |
| *Kale et al. (2009)* | Demyelination, axonal damage | N | U | N | N |
| *Kale, Agaoglu & Tanik (2010)* | Demyelination, axonal damage | N | U | U | N |
| *Kandler et al. (1991)* | Demyelination | N | U | N | N |
| *Leocani et al. (2006)* | Demyelination, axonal damage, remyelination | N | U | U | N |
| *Magistris et al. (1999)* | Demyelination, axonal damage | N | U | N | N |
| *Mayr et al. (1991)* | Demyelination | U | U | N | N |
| *Pisa et al. (2020)* | Demyelination | N | U | U | N |
| *Ravnborg et al. (1992)* | Demyelination | N | U | N | N |
| *Schmierer et al. (2000)* | Demyelination, axonal damage | N | U | N | N |
| *Schmierer et al. (2002)* | Demyelination, axonal damage | N | U | N | N |
| *Tataroglu et al. (2003)* | Demyelination, excitotoxicity, axonal damage | N | U | N | N |

**Note:**
H, high; L, low; N, no; U, uncertain. See ref: (*Bielekova & Martin, 2004*).

Overall, CMCT had the greatest diagnostic performance of the TMS outcomes reviewed (Fig. 3). Two of the eight CMCT studies had low risk of bias (*Beer, Rösler & Hess, 1995*; *Ravnborg et al., 1992*), two had unclear risk of bias (*Cruz-Martínez et al., 2000*; *Magistris et al., 1999*), and four had high risk of bias (*Hess et al., 1987*; *Mayr et al., 1991*; *Schmierer et al., 2000*; *Tataroglu et al., 2003*). Risk of bias was primarily attributable to inappropriate exclusions and using case-control designs. No study could demonstrate biomarker validity, due to not providing detailed data, demonstrating poor sensitivity, or failing to justify use in epidemiologic studies. However, in the high-quality studies, estimated DOR was 6–14, suggesting abnormal CMCT was associated with a large, 6 to 14 times, increased odds of MS (Fig. 3) (*Chen, Cohen & Chen, 2010*).

A major strength of the 2017 McDonald criteria (*Thompson et al., 2018*) is its high sensitivity, owed to using MRI for evidence of lesion dissemination in space and time in persons following a clinical demyelinating episode. In one study, the sensitivity of the 2017

McDonald criteria was as high as 100% (*Gobbin et al., 2019*). CMCT therefore has little additive value to enhance the sensitivity of MS diagnostic criteria. However, the greatest strength of CMCT would be its high specificity (median 100%), where the 2017 McDonald criteria was as low as 14% specific in one study (*Gobbin et al., 2019*). CMCT could reduce false-positive diagnoses (*Schwenkenbecher et al., 2019*), by "ruling-in" persons with corticospinal conduction loss who were identified as having MS as per the 2017 McDonald criteria.

## Alternative utilities for TMS techniques

The current review also found a compelling role for TMS in characterizing MS natural history. For example, CMCT was highly sensitivity for both progressive MS and higher disability status. Given most of the evidence herein is cross-sectional, the authors cannot assign any causal relationship between corticospinal conduction deficits and disease progression or disability accumulation (*Pachner, 2021*). Nevertheless, CMCT could identify early neurodegeneration to help diagnose a transition to SPMS or revise the diagnosis of RMS to PPMS. The potential role of CMCT in identifying occult neurodegeneration would be especially important, given there are limited treatment options for progressive MS and the diagnosis of PPMS or conversion of RMS to SPMS requires evidence of chronic and irreversible disability accumulation (*Hamdy et al., 2022*; *Thompson et al., 2018*). To better establish whether CMCT could expedite the diagnosis of progressive MS subtypes will require more evidence from prospective longitudinal studies.

## Limitations

Despite the novelty of this review in terms of addressing the diagnostic performance of TMS and critically appraising the TMS biomarker literature, several limitations should be noted. First, the 2017 McDonald criteria (*Thompson et al., 2018*) currently represents the gold standard of MS diagnosis. None of the articles reviewed used these criteria, and most used the Poser criteria (*Poser et al., 1983*). While the authors could not identify any past research that directly compared Poser and 2017 McDonald criteria, successive iterations of the McDonald criteria have generally been shown to diagnose MS earlier and more frequently than the Poser criteria (*i.e.*, enhanced sensitivity) (*Brownlee et al., 2015*), but with compromised specificity and a higher rate of misdiagnosis (*Tintoré et al., 2003*). Sensitivity and specificity of the Poser criteria have been estimated at 87% and 94%, respectively (*Engell, 1988*; *Izquierdo et al., 1985*), whereas sensitivity and specificity of the 2017 McDonald criteria range between 68–100% and 14–61%, respectively (*Filippi et al., 2018*, *2022*; *Gobbin et al., 2019*; *van der Vuurst de Vries et al., 2018*). In past work, the addition of CMCT did not enhance the sensitivity of the Poser criteria to increase MS diagnoses, but increased specificity and reduced MS misdiagnoses (*Beer, Rösler & Hess, 1995*). It presently is unclear how TMS techniques would perform in the context of the 2017 McDonald criteria, and this question should be addressed in future research.

Next, this critical review is based on only 17 studies of a select few TMS techniques. Methods like paired-pulse and dual-coil TMS, or TMS-EEG, offer unique ways to explore intracortical excitability in excitatory, inhibitory, and neuromodulatory interneurons;

other regions important to sensorimotor function (*e.g.*, premotor cortex, supplementary motor area, cerebellum, somatosensory cortices); and distant non-motor regions (*Rossini et al., 2015*). Such techniques could characterize biologically plausible disease mechanisms not reviewed here, such as acute excitotoxicity or chronic neurodegeneration, linked to disease activity and progression, respectively (*Chaves et al., 2019*; *Snow et al., 2019*).

Lastly, given this is a critical narrative review, the results must be interpreted more carefully compared to a systematic review or meta-analysis. The studies included in the review are heterogeneous in terms of study design, sample size, participant characteristics, and TMS methods. The current approach to data synthesis and interpretation does not take study heterogeneity into account. Moreover, because the authors did not produce a single estimate for the diagnostic performance of each TMS technique, there is greater onus on the reader to interpret the findings. Nonetheless, the authors provide a protocol-driven review, following evidence-based methods for data extraction and critical appraisal.

## CONCLUSIONS

MS is an immune-mediated neurodegenerative disease characterized by attacks of inflammatory demyelination and axonal damage, with variable but continuous accumulation of disability. Various TMS techniques can characterize conduction loss and axonal damage in the corticospinal tract. Most notably, CMCT could be a putative biomarker to: (1) enhance the specificity of the 2017 McDonald criteria by "ruling-in" true-positive MS diagnoses, (2) revise a diagnosis from RMS to PPMS, or (3) help arrive at an earlier diagnosis of SPMS. Herein, the authors summarized the current state of the literature and determined both a high risk of bias and poor justification for the validity of TMS techniques as diagnostic biomarkers in MS. In the future, more rigorous, prospective, longitudinal studies will be required, using comparisons to the 2017 McDonald criteria.

### Funding
The authors received no funding for this work.

### Competing Interests
Michelle Ploughman is an Academic Editor for PeerJ.

### Author Contributions
- Nicholas J. Snow conceived and designed the experiments, performed the experiments, analyzed the data, prepared figures and/or tables, authored or reviewed drafts of the article, and approved the final draft.
- Hannah M. Murphy conceived and designed the experiments, analyzed the data, authored or reviewed drafts of the article, and approved the final draft.
- Arthur R. Chaves conceived and designed the experiments, analyzed the data, authored or reviewed drafts of the article, and approved the final draft.

- Craig S. Moore conceived and designed the experiments, analyzed the data, authored or reviewed drafts of the article, and approved the final draft.
- Michelle Ploughman conceived and designed the experiments, analyzed the data, authored or reviewed drafts of the article, supervision, Project Administration, and approved the final draft.

## Data Availability

This is a literature review. Additional data can be found in the Supplemental Files.

## Supplemental Information

Supplemental information for this article can be found online at http://dx.doi.org/10.7717/peerj.17155#supplemental-information.

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
