# Peer review of "Transcranial magnetic stimulation enhances the specificity of multiple sclerosis diagnostic criteria: a critical narrative review"

_PeerJ, doi:10.7717/peerj.17155_

## Round 0.1 · original submission · Major Revisions

Please revise the manuscript with clarity the comments of the authors.

Reviewer 1 ·

Basic reporting

Overall, the message of this review is clear. However, there are some concerns that the authors need to address.

1) Introduction:
(a) Background information is inadequate and not supported with current literature. E.g., lines 65-68.
b) The authors tend to use a monoreferential style in a review paper, which should be avoided. E.g., lines 82-87
(c) The authors need to specify the TMS measures mentioned in lines 77-79
d) It is still not clear what the problem is with the existing diagnosis and why MS patients need the alternative diagnosis - see line 82

2) Methodology
a) Inadequate description of search strategy. Authors need to describe who performed this search
b) In data extraction, authors need to describe who and how transcription of 2x2 contigency findings was performed and what happened to the selected study if sensitivity and specificity results were not available
(c) The description of the critical appraisal is not sufficient. The authors must indicate who performed this assessment. The explanation of biomarker validity testing is not sufficient.

3) Results
(a) Lack of clarity in the write up. Authors did not tell the reader which table/data to refer to (see line 177 - 202)
b) Line 204: The authors state the imaging results in the title. However, this paragraph does not provide information on EEG, fMRI, and other neuroimaging modalities.
c) Lines 204-216: The authors did not tell the reader which table/data to refer to, i.e., there is a lack of clarity. In fact, no correlation values are given in the table. What association the authors are trying to imply here is still unclear.

4) Discussion
a) Mixing the study designs of the selected articles could also be a limitation. The authors might consider including this as a limitation?

5) Figure 3: The authors did not clearly explain how they arrived at this forest diagram. This should also be explained in the method

Experimental design

as stated above

Validity of the findings

as stated above

Additional comments

as stated above

Reviewer 2 ·

Basic reporting

This article was written clearly.The main issue was clearly highlighted and well explained with strong arguments based on previous research.
References were balance and up to date.
Information in this article was profesionally structured, and raw data were shared.

Experimental design

Rigorous investigation was performed and achieved a high technical and ethical standard.

Method was sufficiently described.

Validity of the findings

Findings were arranged in a systematic way to convey the information. Conclusion was also well stated, linked to original research question.

---

## Round 0.2 · accepted · Accept

Thank you the manuscript has been accepted.

Reviewer 1 ·

Basic reporting

The authors improved their manuscripts and responded to my comments satisfactorily.

Experimental design

The authors improved their manuscripts and responded to my comments satisfactorily.

Validity of the findings

The authors improved their manuscripts and responded to my comments satisfactorily.